# A remote-controlled automatic chest compression device capable of moving compression position during CPR: A pilot study in a mannequin and a swine model of cardiac arrest

Gil Joon Suh[1,2,3]*, Taegyun Kim[2], Kyung Su Kim[2,3], Woon Yong Kwon[1,2,3], Hayoung Kim[2], Heesu Park[2], Gaonsorae Wang[2], Jaeheung Park[4,5], Sungmoon Hur[5], Jaehoon Sim[4], Kyunghwan Kim[6], Jung Chan Lee[3,7,8,9,10], Dong Ah Shin[7], Woo Sang Cho[8], Byung Jun Kim[8], Soyoon Kwon[8], Ye Ji Lee[11]

1 Department of Emergency Medicine, Seoul National University College of Medicine, Seoul, Republic of Korea, 2 Department of Emergency Medicine, Seoul National University Hospital, Seoul, Republic of Korea, 3 Research Center for Disaster Medicine, Seoul National University Medical Research Center, Seoul, Republic of Korea, 4 Graduate School of Convergence Science and Technology, Seoul National University, Seoul, Republic of Korea, 5 Advanced Institutes of Convergence Technology, Suwon, Republic of Korea, 6 NT Robot, Co, Seoul, Republic of Korea, 7 Institute of Medical and Biological Engineering, Medical Research Center, Seoul National University, Seoul, Republic of Korea, 8 Interdisciplinary Program in Bioengineering, Graduate School, Seoul National University, Seoul, Republic of Korea, 9 Department of Biomedical Engineering, Seoul National University College of Medicine, Seoul, Republic of Korea, 10 Department of Biomedical Engineering and Innovative Medical Technology Research Institute, Seoul National University Hospital, Seoul, Republic of Korea, 11 Biomedical Research Institute, Seoul National University Hospital, Seoul, Republic of Korea

* suhgil@snu.ac.kr

**Data Availability Statement:** All files (baseline. xlsx, data_open.csv) are available from https://doi.org/10.34740/KAGGLE/DSV/6304355.

## Abstract

### Background

Recently, we developed a chest compression device that can move the chest compression position without interruption during CPR and be remotely controlled to minimize rescuer exposure to infectious diseases. The purpose of this study was to compare its performance with conventional mechanical CPR device in a mannequin and a swine model of cardiac arrest.

### Materials and methods

A prototype of a remote-controlled automatic chest compression device (ROSCER) that can change the chest compression position without interruption during CPR was developed, and its performance was compared with LUCAS 3 in a mannequin and a swine model of cardiac arrest. In a swine model of cardiac arrest, 16 male pigs were randomly assigned into the two groups, ROSCER CPR (n = 8) and LUCAS 3 CPR (n = 8), respectively. During 5 minutes of CPR, hemodynamic parameters including aortic pressure, right atrial pressure, coronary perfusion pressure, common carotid blood flow, and end-tidal carbon dioxide partial pressure were measured.

**Funding:** This research was supported by a grant of the Korea Health Technology R&D Project through the Korea Health Industry Development Institute (KHIDI), funded by the Ministry of Health & Welfare, Republic of Korea (grant number: HW20C2132). GJS received this fund. The funders had no role in study design, data collection and analysis, decision to publish, or preparation of the manuscript.

**Competing interests:** The authors have declared that no competing interests exist.

## Results

In the compression performance test using a mannequin, compression depth, compression time, decompression time, and plateau time were almost equal between ROSCER and LUCAS 3. In a swine model of cardiac arrest, coronary perfusion pressure showed no difference between the two groups ($p = 0.409$). Systolic aortic pressure and carotid blood flow were higher in the LUCAS 3 group than in the ROSCER group during 5 minutes of CPR ($p < 0.001$, $p = 0.008$, respectively). End-tidal $CO_2$ level of the ROSCER group was initially lower than that of the LUCAS 3 group, but was higher over time ($p = 0.022$). A Kaplan-Meier survival analysis for ROSC also showed no difference between the two groups ($p = 0.46$).

## Conclusion

The prototype of a remote-controlled automated chest compression device can move the chest compression position without interruption during CPR. In a mannequin and a swine model of cardiac arrest, the device showed no inferior performance to a conventional mechanical CPR device.

## Introduction

Cardiac arrest is one of the leading causes of death, and despite advances in resuscitation, the survival to discharge rate is only about 10% [1]. Current guidelines recommend that high-quality cardiopulmonary resuscitation (CPR), the most important lifesaving intervention for the survival of a patient in cardiac arrest, should include chest compressions to the lower half of the sternum to a depth of at least 5 cm and at a rate of 100 to 120/min while minimizing pauses in compressions [2]. However, high-quality manual CPR is extremely exhausting for rescuers, which easily causes rescuer's fatigue [3–6]. To solve these problems in manual CPR, several mechanical CPR devices such as the Autopulse (Zoll Medical Co., MA, USA) and LUCAS (Physio-Control Inc., WA, USA) have been developed to provide CPR without interruption [7, 8]. However, these devices are fixed to the chest and only compress on one position, so they must be unbound when the compression position needs to be changed. This unbinding may result in interruption of chest compressions, compromising CPR quality [9]. During the COVID-19 pandemic, it is important for health care providers to protect themselves and their colleagues from unnecessary exposure to COVID-19 during CPR. In 2020, the American Heart Association, along with its collaborating organizations issued "Interim Guidance to Health Care Providers for Basic and Advanced Cardiac Life Support in Adults, Children, and Neonates with Suspected or Confirmed COVID-19", which has been updated annually. The guidance recommends considering the use of mechanical CPR devices for adults and adolescents to reduce the number of rescuers and rescuers exposure in settings with protocols in place and expertise in their use [10–12]. The European Resuscitation Council and the 2020 Korean Cardiopulmonary Resuscitation Guidelines also recommend the use of mechanical CPR to minimize contact with the patient [13, 14]. However, existing mechanical CPR devices still pose a risk of exposure to COVID-19 because rescuers must perform all of the operating processes, such as installing them on a patient and driving them next to the patient. In order to solve these problems, mechanical CPR device must be able to change the compression position during CPR and be remotely controlled to minimize exposure of rescuers to infectious diseases including COVID-19.

Recently, we developed a remotely controlled automatic chest compression device capable of moving the chest compression position during CPR. The purpose of this study was to compare its performance with conventional mechanical CPR device in a swine model of cardiac arrest.

## Materials and methods

### Development of a prototype of remote-controlled automatic chest compression device capable of moving chest compression position during CPR

We developed a prototype of remote-controlled automatic chest compression device that can move the position of chest compression during CPR, and named it ROSCER (S1 Table). ROSCER consists of a hood, a compression unit, a supporting unit, and a user control panel. The hood includes a controller, three BLDC motors, three motor drivers, and a battery. The supporting unit consists of a plastic back plate and two support legs (Fig 1A). The user control panel is situated on the hood and is connected to the controller. The controller can control ROSCER's chest compression rate, depth, and position remotely through a cable about 10 meters away (Fig 1B).

The hood includes a controller, three BLDC motors, three motor drivers, and a battery. Three motors are used for generating horizontal (X, Y axis) and vertical (Z axis) movement. The compression unit consists of a piston and an actuator and functions to compress the chest. The compression unit is connected to three motors to allow for horizontal repositioning of compressions without interfering chest compressions (Fig 2A and 2B). The user control panel is the user interface with which the device can be controlled through sixteen button switches and two LED's for warning and battery status. The compression rate can be adjusted to 90, 100, 110 or 120 per minute, and compression depth ranges from 4 to 6. The controller can take over its control function to the remote control pendant by pressing mute button for five seconds. The controller communicates with a remote control pendant through RS485 communication (Fig 2C and 2D).

### Compression performance test using a mannequin

In the performance test using a mannequin, the ROSCER's compression force obtained through the force sensor, compression depth, and motion range of compression position change were measured. A performance comparison experiment between ROSCER and LUCAS 3 (Stryker, MI USA) was conducted.

### Pilot study in a swine model of cardiac arrest

In order to evaluate the performance of simple chest compression except for the ROSCER's ability to change the position and rate of chest compressions, a comparative experiment with LUCAS 3 was performed in a swine cardiac arrest model. This study was conducted in accordance with the Animal Research: Reporting of *In Vivo* Experiments (ARRIVE) guidelines 2.0 [15]. All experiments were approved by the Institutional Animal Care and Use Committee in Seoul National University Hospital (SNUH-IACUC No. 21-0035-C1A0) and animals were maintained in the facility accredited AAALAC International (#001169) in accordance with Guide for the Care and Use of Laboratory Animals 8[th] edition, NRC (2010) [16].

**Swine model of cardiac arrest.** A swine cardiac arrest model was developed in our previous experimental study [17]. In brief, the experiments were carried out on 16 male pigs which were conventional mixed-breed male pigs from Landrace-Yorkshire with median age of 15

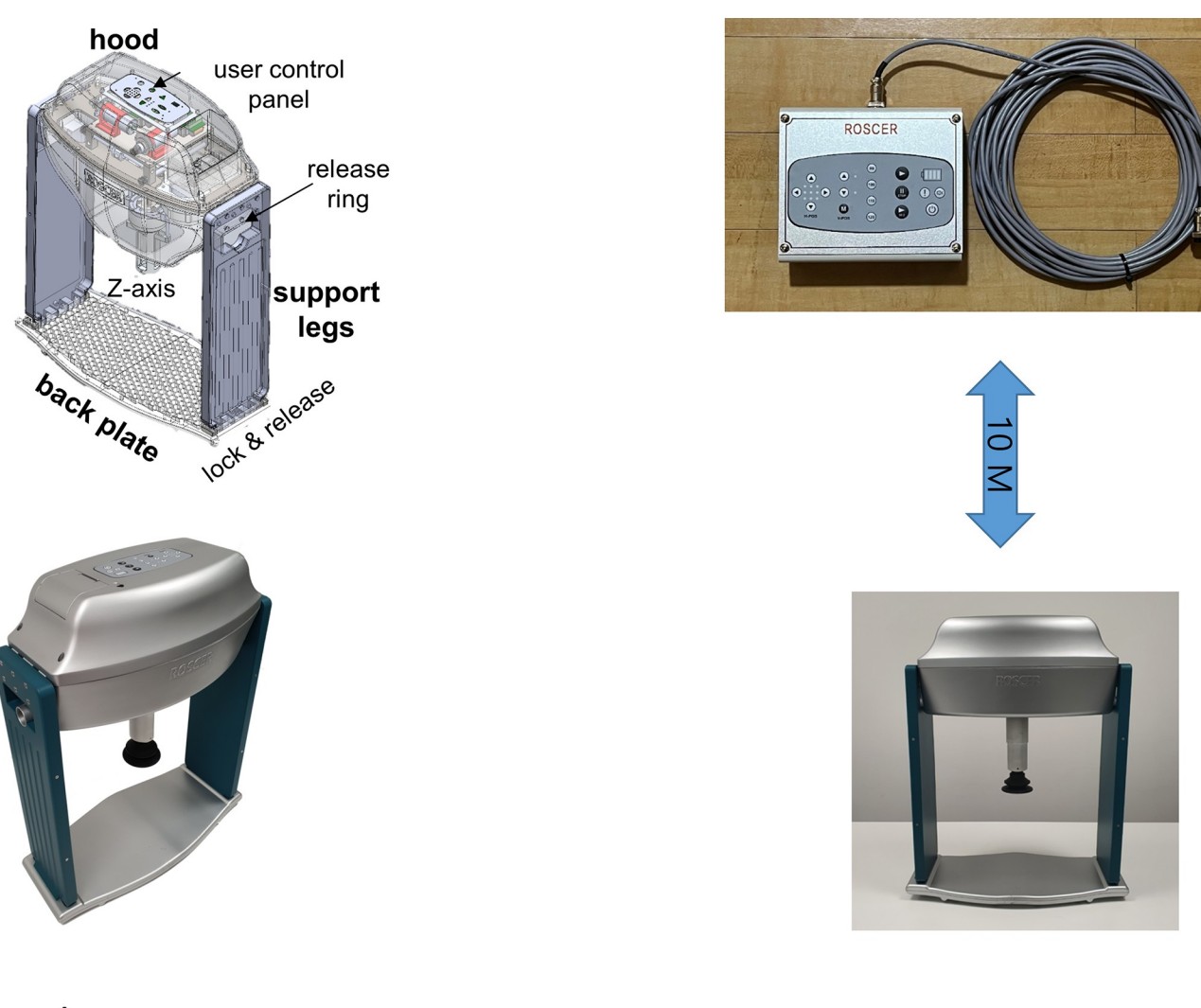

**Fig 1. Configuration of ROSCER.** Reprinted from under a CC BY license, with permission from [NT Robot, Co], original copyright [2023].

weeks (range, 14–19 weeks) and median weight of 45 kg (range 40–50 kg). We chose pigs as experimental animals because pigs and humans are physiologically very similar in size and shape of their heart, and we have several experiences about the cardiac arrest and CPR in Landrace pig species. Experimental animals were fed twice a day. The air temperature of the breeding rooms was maintained in the range of 18°C to 29°C with 10 and 14 hours of light and dark exposure, respectively. The animals underwent an acclimatization period of 14 days before the experiments. After the induction of anaesthesia with intramuscular Zoletil (zolazepam and tiletamine, 5 mg/kg; Virbac AH, Fort Worth, TX) and followed by inhalation of 1% isoflurane, experimental animals were mechanically ventilated using a ventilator (GE Datex-Ohmeda S/5 Aespire Anesthesia Machine, Buckinghamshire, UK) after inserting a 6.5F endotracheal tube. The initial mechanical ventilation settings were adjusted to apply a tidal volume of 10 mL/kg and a frequency of 15/min and maintain the end-tidal $CO_2$ level within 35 to 40 mmHg. Lactated Ringer's solution was administered through the ear vein at a rate of 4 mL/kg/hr. After the right common carotid artery and internal jugular vein were exposed, 8.5F sheath

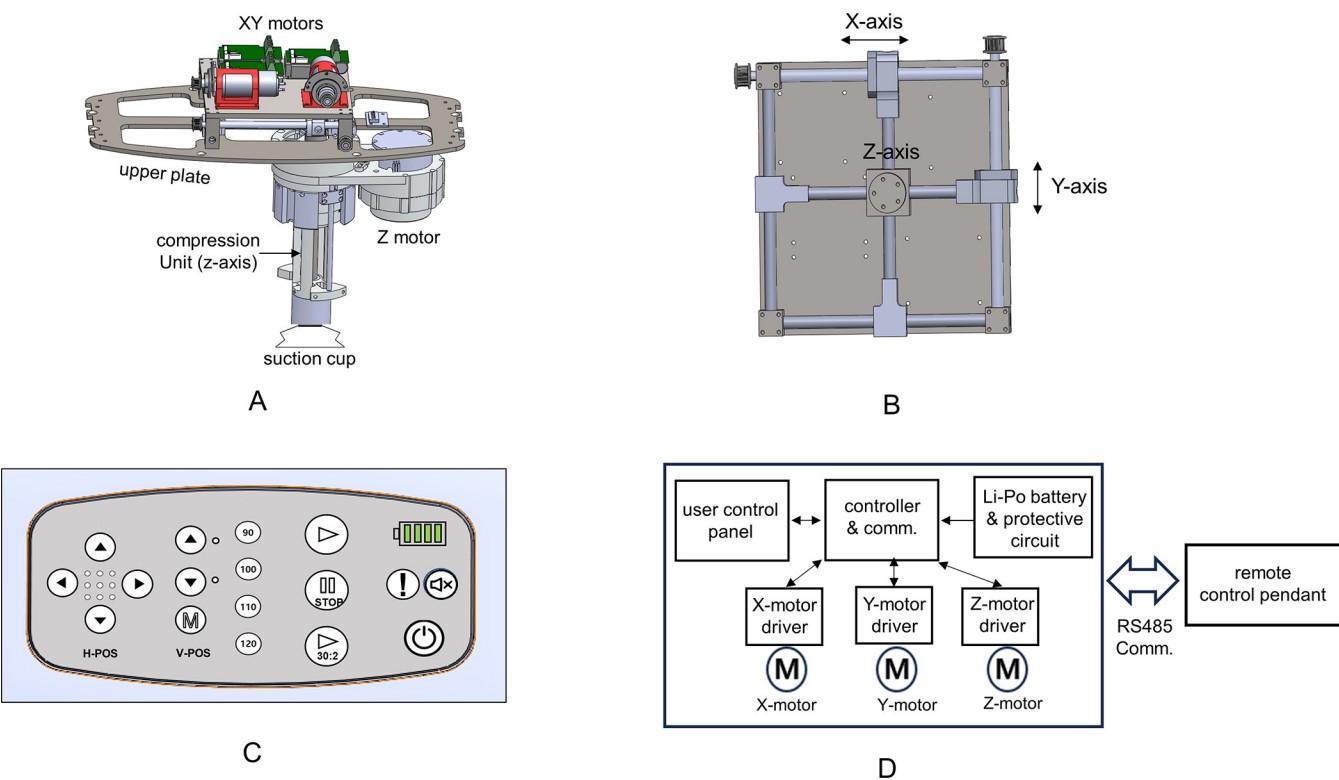

**Fig 2. Mechanism for horizontal and vertical movement and the user control panel of ROSCER.** A: mechanism for XYZ movement; B: mechanism under upper plate; C: user control panel; D: control block diagram of ROSCER. Reprinted from under a CC BY license, with permission from [NT Robot, Co], original copyright [2023].

introducers (Arrow International, Cleveland, OH) were cannulated into the artery and vein. The left common carotid artery was exposed and a perivascular Flowprobe (3PSB or 4PSB according to the artery diameter; Transonic Systems Inc, NY USA) was applied around the left common carotid artery to measure carotid blood flow. After stabilizing the animals for 15 min, a pacing catheter was placed in the right ventricular wall via the right internal jugular vein introducer to induce ventricular fibrillation (VF).

**Experimental protocol.** To compare the effect of simple chest compression of ROSCER with LUCAS 3, 16 pigs were randomly assigned into the two groups using a research randomizer (version 4.0, http/www.randomizer.org/), ROSCER CPR (n = 8) and LUCAS 3 CPR (n = 8), respectively. After the baseline data measurement, VF was induced by passing a direct current of 9 V for 5 seconds. Cardiac arrest was confirmed by VF waveform on the ECG and mean arterial pressure less than 15 mmHg. After 7 minutes of no-flow time, CPR using mechanical CPR device (5 cm depth, 100 beats per minute) with artificial ventilation (30:2 of compression-to-ventilation ratio) was initiated. After 5 minutes of CPR, transthoracic defibrillation (biphasic, 150 J) was performed every 2 minutes using a Zoll R Series Defibrillator (Zoll Medical, Chelmsford, MA). After 8 minutes of CPR, the animals were given 1 mg of adrenaline (epinephrine, IV) every 3 minutes if return of spontaneous circulation (ROSC) was not achieved. ROSC was defined as maintenance of a systolic arterial blood pressure of at least 60 mmHg for at least 10 consecutive minutes. After 20 minutes of ROSC, the animals were euthanized using 20 mEq of KCl. When ROSC was not achieved despite 25 minutes of CPR, the animal was terminated (Fig 3).

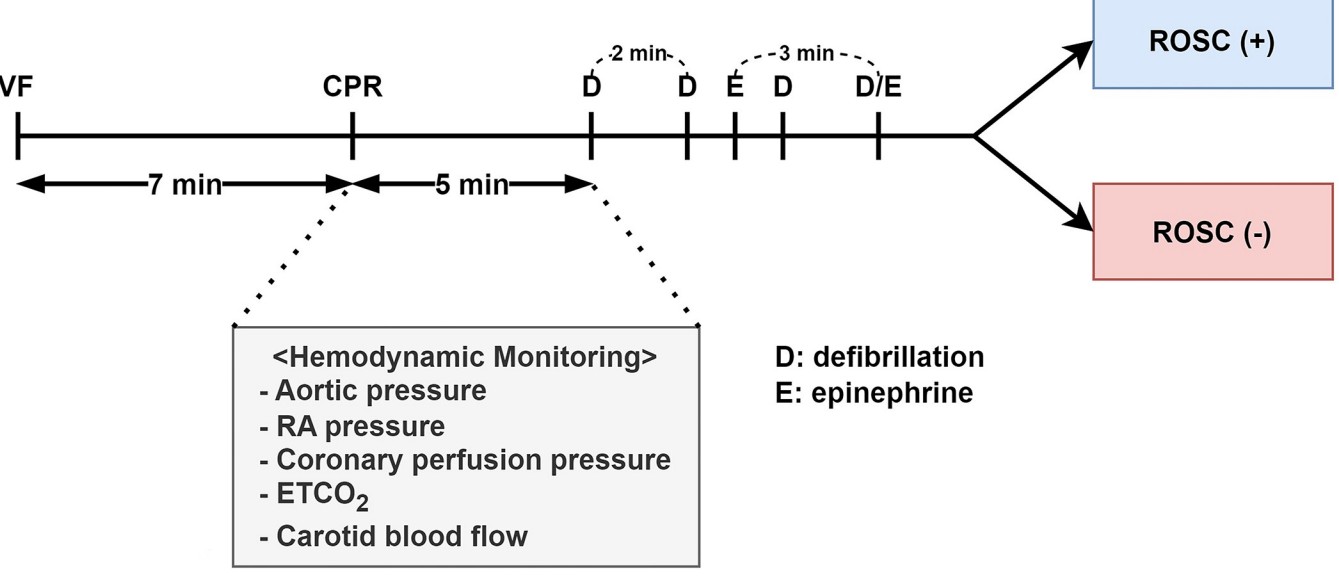

**Fig 3. Schematic diagram of experiment in a swine model of cardiac arrest.**

During 5 minutes of CPR, hemodynamic parameters including aortic pressure, right atrial pressure, coronary perfusion pressure (CPP), common carotid blood flow, and end-tidal carbon dioxide partial pressure (ETCO$_2$) were measured. CPP was defined as the difference between aortic pressure and right atrial pressure during the diastolic (decompression) phase of mechanical CPR [18]. CPP was measured using mid-diastolic method in which right atrial blood pressure was subtracted from time-coincident aortic blood pressure at the midpoint of the diastolic phase [19]. A Kaplan-Meier survival analysis for ROSC was conducted.

## Statistical analysis

For baseline characteristics and CPR outcomes, data were presented as means ± standard deviations and were compared using Student's t-tests. Linear mixed models were used to analyze hemodynamic variables including CPP, aortic pressure, right atrial pressure and ETCO$_2$ levels. Variables included in each linear mixed models were time divided into 10-second intervals, experimental group and group-time interaction. A Kaplan-Meier analysis with a log-rank test was performed to compare the ROSC rate between the two groups. Two-sided p values < 0.05 were regarded as statistically significant. LabChart 8 with (ADInstruments, Dunedin, New Zealand) and R version 4.3.0 (R foundation) were used for peak detection and all other analyses, respectively.

## Results

### Compression performance test using a mannequin

In the compression performance test using a mannequin, the compression force of ROSCER obtained through the force sensor was 500N or more, and the compression depth was consistently 5 cm (Fig 4A). A performance comparison experiment between ROSCER and LUCAS 3 (Stryker, MI USA) was conducted. It was confirmed that the one cycle and duty cycle of the waveforms of the two devices was the same at about 0.6 seconds and 50%, respectively, and that the compression depth, compression time, decompression time, and plateau time were almost equal (Fig 4B). The range of motion for changing the compression position was within

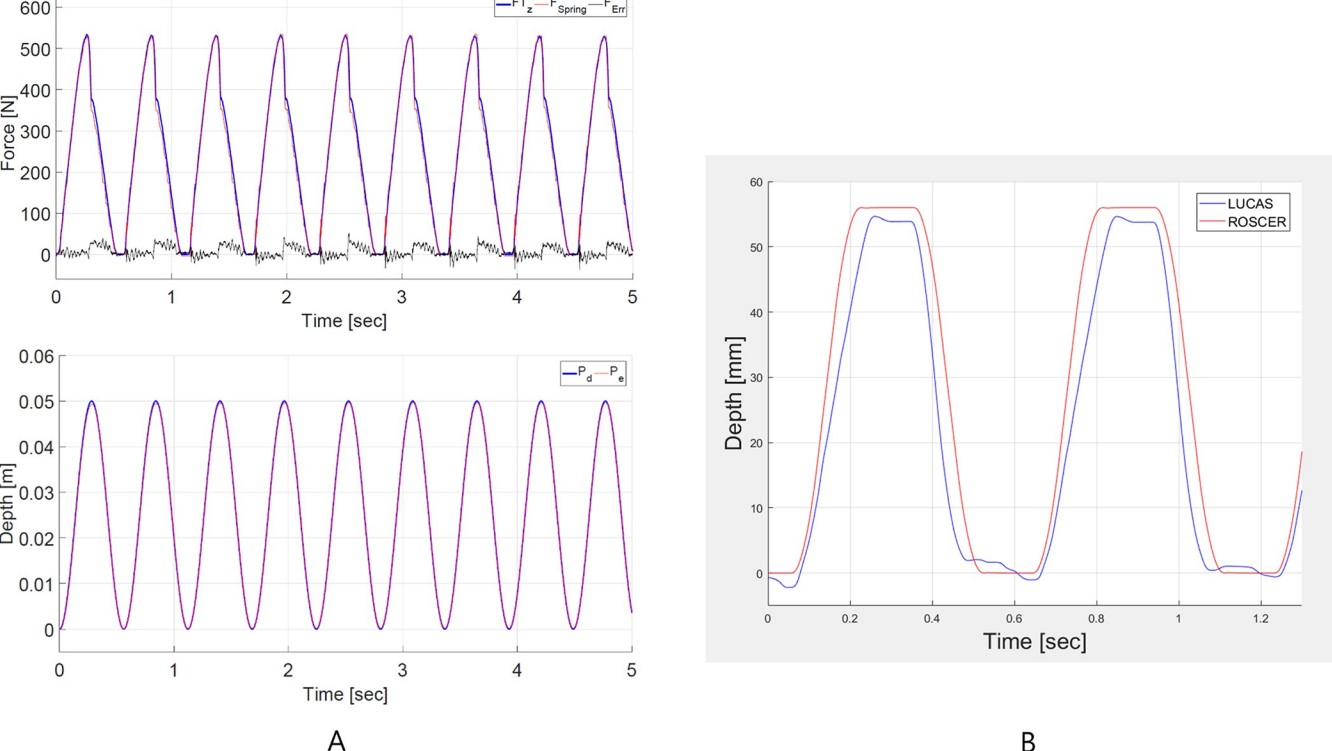

**Fig 4. Compression performance test of ROSCER using a mannequin.** The compression force and depth of ROSCER (A). A performance comparison test using a mannequin between ROSCER and LUCAS 3 (Stryker, MI USA) (B).

1 cm to 3 cm (S1 and S2 Videos). In addition, the time taken for a total of 8 participants, 2 per group, to install and operate ROSCER and LUCAS 3 was 16.5 [16.0–17.5] seconds and 18.5 [17.0–23.5] seconds ($p = 0.234$), respectively.

## Comparison of ROSCER with LUCAS 3 in a swine model of cardiac arrest

Prior to the induction of cardiac arrest, baseline characteristics including body weight, systolic aortic pressure, diastolic aortic pressure, mean aortic pressure, right atrial pressure, peak carotid blood flow, mean carotid blood flow, heart rate, and $ETCO_2$ between ROSCER and LUCAS 3 were not significantly different (Table 1).

There was no difference in CPP between the two groups ($p = 0.409$) (Fig 5A). Mean carotid blood flow at each time points was higher in the LUCAS 3 group than in the ROSCER group ($p = 0.008$) (Fig 5B). The $ETCO_2$ value of the ROSCER group was initially lower than that of the LUCAS 3 group, but was higher over time ($p = 0.022$) (Fig 5C). Aortic pressure and right atrial pressure at the peak of systolic phase (compression phase) were higher in the LUCAS 3 group than in the ROSCER group during 5 minutes of CPR ($p < 0.001$ and $p < 0.001$, respectively). There was also no difference in Aortic pressure, and right atrial pressure at the midpoint of the diastolic phase (decompression phase) between the two groups ($p = 0.213$, and 0.113, respectively) (S1 Fig). A Kaplan-Meier survival analysis for ROSC also showed no difference between the two groups ($p = 0.46$) (Fig 5D). The time to ROSC in the ROSCER and LUCAS 3 groups was 7.0 (5.0–9.0) min and 9.0 (9.0–12.0) min, respectively ($p = 0.167$). Mean waveforms of aortic pressure, right atrial pressure, and CPP at 0, 1, 2, 3, 4, and 5 minutes were also analyzed (S2 Fig).

**Table 1. Baseline characteristics.**

|  | LUCAS 3 (n = 8) | ROSCER (n = 8) | *p* value |
|---|---|---|---|
| Body weight, kg | 43.1 ± 1.6 | 45.8 ± 5.0 | 0.186 |
| Systolic aortic pressure, mmHg | 109.4 ± 7.6 | 115.5 ± 11.4 | 0.227 |
| Diastolic aortic pressure, mmHg | 84.8 ± 4.9 | 88.4 ± 8.8 | 0.325 |
| Mean aortic pressure, mmHg | 97.5 ± 6.3 | 103.0 ± 10.2 | 0.216 |
| Right atrial pressure, mmHg | 4.4 ± 2.7 | 5.0 ± 3.9 | 0.717 |
| Peak carotid blood flow, mL/min | 542.3 ± 72.1 | 579.6 ± 61.3 | 0.284 |
| Mean carotid blood flow, mL/min | 346.4 ± 65.7 | 401.3 ± 58.2 | 0.099 |
| Heart rate, beats per min | 127.2 ± 32.4 | 141.5 ± 25.6 | 0.346 |
| ETCO$_2$, mmHg | 40.6 ± 3.8 | 43.6 ± 4.7 | 0.182 |

ETCO$_2$: end-tidal carbon dioxide partial pressure

## Discussion

We developed a prototype of a remote controlled automatic chest compression device that can change compression position (ROSCER). In our simulation experiment, ROSCER showed equivalent compression depth and compression force trajectories compared to LUCAS 3. In a swine model of cardiac arrest, CPP and ROSC rate were not significant between the two groups during 5 minutes of CPR. The ETCO$_2$ value was initially lower than that of the LUCAS 3 group, but was higher over time. However, aortic pressure and right atrial pressure at the peak of systolic phase, and mean carotid blood flow were significantly higher in the LUCAS 3 group than in the ROSCER group.

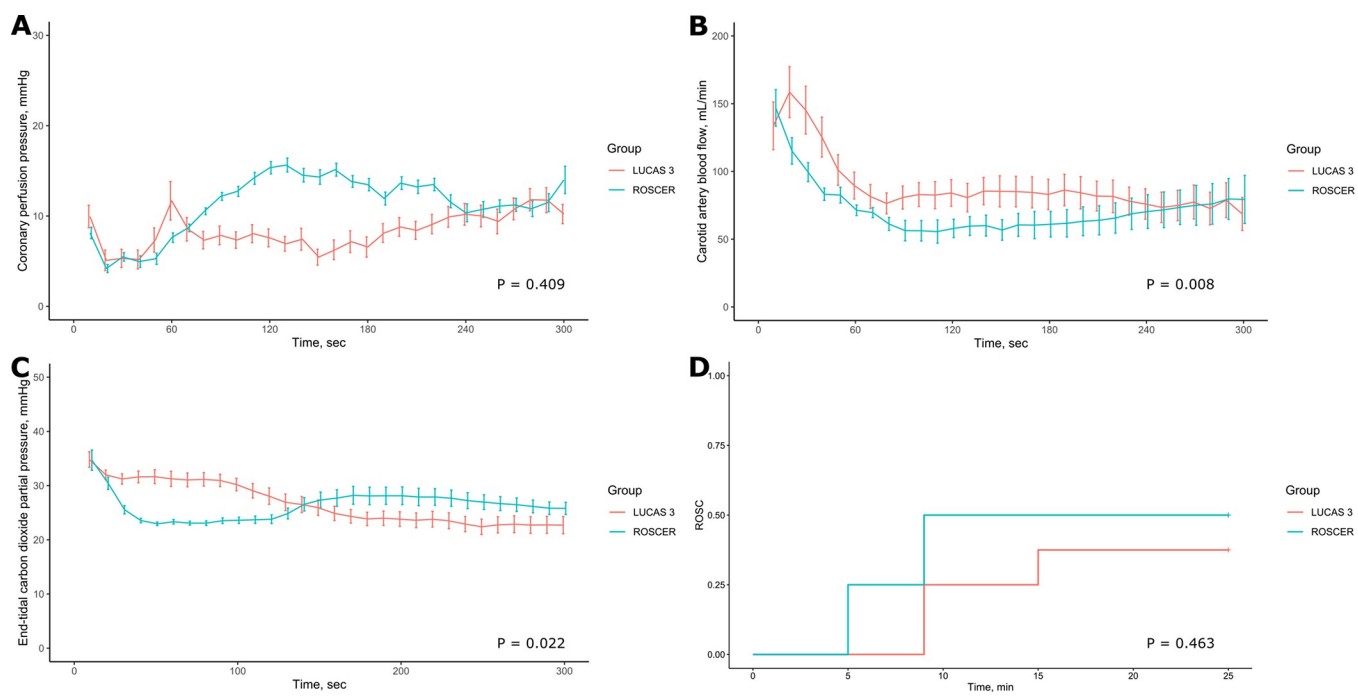

**Fig 5. Hemodynamic measurements and ROSC rate during CPR.** Coronary perfusion pressure (A). Carotid blood flow (B). The End-tidal CO$_2$ level (C). A Kaplan-Meier survival analysis for ROSC (D). Points and error bars represent means ± standard errors.

CPP is a surrogate for myocardial blood flow when the coronary vascular resistance is approximately constant and is a major indicator of the effectiveness of CPR and ROSC [20]. CPP achieved by standard closed chest CPR is typically reported as 10–20 mmHg [18, 19]. A clinical study reported that a CPP threshold of 15 mmHg was required for ROSC [21]. The reason why CPP was slightly higher in the ROSCER group than in the LUCAS 3 group may be related to the greater difference in aortic pressure (about 20 mmHg) than the difference in RAP (about 5 mmHg) in the mid-diastolic phase (S1 Fig). In this study, CPP was measured using the mid-diastolic method, which subtracts the right atrial pressure from the time-matched aortic pressure at the midpoint of the diastolic (decompressive) phase. Various methods of measuring CPP are known, and differences may occur depending on the measurement method [19]. Therefore, the CPP value measured in this study may differ from the actual value.

In this study, stroke volume was not measured directly, but instead, common carotid blood flow was measured as an indirect surrogate of stroke volume. The higher common carotid blood flow in LUCAS 3 compared to ROSCER during compression may be related to an increase in stroke volume, which can cause an increase in aortic pressure. Based on the chest pump theory, stroke volume blood flow during CPR is related to an increase in intrathoracic pressure generated by sternum compression, rather than direct compression of the heart itself. Increased intrathoracic pressure also leads to an increase in right atrial pressure [22, 23]. Higher aortic pressure, right atrial pressure, and carotid blood flow in LUCAS 3 compared to ROSCER during compression may be related to increased intrathoracic pressure due to higher compression force. In the simulation experiment using a mannequin, the compression profiles of the two devices were almost the same. However, in the swine model of cardiac arrest, these hemodynamic differences appear to be the result of the ROSCER's inability to generate sufficient compression force. In terms of structure, the size of ROSCER is slightly different from that of LUCAS 3. The dimensions of ROSCER are 620 (H) x 547 (W) x 245 (D) mm and LUCAS 3 are 560 (H) x 520 (W) x 240 (D) mm, respectively. ROSCER was designed to be slightly larger than LUCAS 3 in height and width to accommodate larger patients. This structure may cause deformation of both supporting legs during chest compressions, resulting in a somewhat unstable state, which may result in insufficient compression. In particular, because the pig's chest is V-shaped, if the supporting legs become unstable due to deformation, there is an increased possibility of slipping during compression, which may result in insufficient chest compression, which leads to lower aortic pressure or carotid artery pressure. In future research, we will consider ways to minimize deformation by increasing the rigidity of the support leg material.

A complete chest recoil is one of the important factors for high-quality CPR. Incomplete chest wall recoil is associated with increased intrathoracic pressure and decreased coronary perfusion. The 2020 AHA guidelines recommend that rescuers avoid leaning on the chest between compressions to allow complete chest wall recoil for adults in cardiac arrest [2]. In a cadaver study, the elastic recoil of the human thorax decreases over time during CPR, which could negatively affect the heart refilling. In the study, the anterior posterior chest diameter difference of roughly 1.5 cm between the start and the end of CPR represents the chest collapse produced by CPR using a manual ACD-CPR device [24]. Therefore, mechanical CPR requires active decompression, such as piston-mounted suction cups or modified ACD-CPR devices, to compensate for reduced chest recoil during CPR [25, 26].

The strength of the ROSCER device is that the compression position can be changed during chest compressions. However, we did not conduct experiment to compare this advantage of the ROSCER device with LUCAS 3 because LUCAS 3 do not have a function to change compression position. Regarding the compression position during CPR, current guidelines

recommend compressing the lower half of the sternum, but a study have reported that the optimal chest compression position differs depending on the patient's body type or gender [27]. Our research team also recently developed the algorithm of the robot CPR system which automatically finds the optimal compression position under the guidance of $ETCO_2$ feedback in swine models of cardiac arrest, and showed good neurological outcome in a comparative experiment with LUCAS CPR [9]. The ROSCER device also can move the position of the chest compression without stopping compression when a change in the compression position is necessary during CPR, minimizing the chest compression pause for high-quality CPR. Another advantage of the ROSCER device can be controlled remotely. With the COVID-19 pandemic, protecting healthcare workers from infection during CPR has become a critical issue. This device is highly valued in that it can be operated remotely through a wired connector of about 10 meters, allowing medical staff to resuscitate patients with suspected infection away from them. A wireless, remote controlled mechanical chest compression device will be developed soon. However, although this remotely controlled automatic chest compression device is designed to minimize exposure to infection in CPR rescuers, this device alone cannot avoid potential exposure to infection from the airway and ventilation during CPR. Therefore, all rescuers should wear appropriate PPE including a respirator, gown, gloves, and eye protection for patients with suspected or confirmed infection when performing intubation and bag-valve mask ventilation. HEPA filtered ventilation must also be provided. Recently, many studies have been reported using mechanical ventilators after a definite airway is secured, so the use of such equipment may also be considered to protect rescuers from infection.

There were several limitations to our study. First, this study is an experimental study conducted in a swine cardiac arrest model. Anatomically, the pig's chest is significantly different from the human chest, and physiologically, there are many differences between a healthy swine cardiac arrest model and a cardiac arrest patient, so much more data and safety and efficacy evaluations of the ROSCER device are needed prior to clinical application. Second, we measured CPP, not myocardial blood flow directly. Cardiac arrest patients have cardiovascular diseases such as atherosclerosis, which may increase coronary vascular resistance, resulting in a decrease in coronary blood flow without a change in CPP. Third, the persons performing the study was not blinded to which device was used as this was experimental study comparing two different devices that could be reliably distinguished by the naked eye. In most research using medical devices, blindness is not possible due to the nature of the medical devices. However, if blinding is not easy, efforts must be made to maintain researcher ethics, and experimental procedures and evaluations must be independent and standardized to minimize evaluation bias. Lastly, the number of animals to assign to each CPR group was too small to show the statistical significance between the ROSCER and LUCAS 3 devices.

## Conclusions

The prototype of a remote-controlled automated chest compression device can move the chest compression position without interruption during CPR. In a swine model of cardiac arrest, the device showed lower systolic aortic pressure and carotid blood flow compared to the LUCAS 3 device, but did not differ in coronary perfusion pressure and ROSC rate. The $ETCO_2$ level was initially lower, but was higher over time.

## Supporting information

**S1 Table. Specifications of ROSCER.**
(DOCX)

**S1 Fig.** Aortic pressure, and right atrial pressure at the systolic (C, D) and mid-diastolic phase (A, B). Points and error bars represent means ± standard errors.
(TIFF)

**S2 Fig. Mean waveforms of aortic pressure, right atrial pressure, and coronary perfusion pressure at 0, 1, 2, 3, 4, and 5 minutes.**
(TIFF)

**S1 Video. The performance test of changing the compression position using a mannequin.**
(MP4)

**S2 Video. The performance test of changing the compression position and remote controlling in a swine model of cardiac arrest.**
(MP4)

# Acknowledgments

We thank the medical research cooperation center (MRCC) for their assistance in the statistical analysis of this study.

# Author Contributions

**Conceptualization:** Gil Joon Suh, Taegyun Kim, Kyung Su Kim, Woon Yong Kwon, Jaeheung Park, Sungmoon Hur, Kyunghwan Kim, Jung Chan Lee.

**Data curation:** Taegyun Kim, Kyung Su Kim, Hayoung Kim, Heesu Park, Gaonsorae Wang, Kyunghwan Kim, Dong Ah Shin, Byung Jun Kim, Soyoon Kwon, Ye Ji Lee.

**Formal analysis:** Gil Joon Suh, Taegyun Kim, Kyung Su Kim, Woon Yong Kwon, Jaeheung Park, Sungmoon Hur, Jaehoon Sim, Jung Chan Lee, Dong Ah Shin, Woo Sang Cho, Byung Jun Kim.

**Supervision:** Gil Joon Suh.

**Writing – original draft:** Gil Joon Suh.

**Writing – review & editing:** Gil Joon Suh, Taegyun Kim.

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
