## [Decision Letter · Decision Letter 0]

4 Dec 2023

PONE-D-23-26031A Remote-controlled automatic chest compression device capable of moving compression position during CPR: A pilot study in a swine model of cardiac arrestPLOS ONE

Dear Dr. Suh,

Thank you for submitting your manuscript to PLOS ONE. After careful consideration, we feel that it has merit but does not fully meet PLOS ONE’s publication criteria as it currently stands. Therefore, we invite you to submit a revised version of the manuscript that addresses the points raised during the review process.

We look forward to receiving your revised manuscript.

Kind regards,

Chiara Lazzeri

Academic Editor

PLOS ONE

https://c.coek.info/pdf-end-tidal-co2-guided-automated-robot-cpr-system-in-the-pig-preliminary-communica.html

In your revision ensure you cite all your sources (including your own works), and quote or rephrase any duplicated text outside the methods section. Further consideration is dependent on these concerns being addressed.

4. We note that Figure(s) 1 and 2 in your submission contain copyrighted images. All PLOS content is published under the Creative Commons Attribution License (CC BY 4.0), which means that the manuscript, images, and Supporting Information files will be freely available online, and any third party is permitted to access, download, copy, distribute, and use these materials in any way, even commercially, with proper attribution. For more information, see our copyright guidelines: http://journals.plos.org/plosone/s/licenses-and-copyright.

a. You may seek permission from the original copyright holder of Figure(s) 1 and 2 to publish the content specifically under the CC BY 4.0 license. 

Reviewers' comments:

Reviewer's Responses to Questions

**Comments to the Author**

1. Is the manuscript technically sound, and do the data support the conclusions?

Reviewer #1: Yes

Reviewer #2: Partly

2. Has the statistical analysis been performed appropriately and rigorously? 

Reviewer #1: Yes

Reviewer #2: Yes

3. Have the authors made all data underlying the findings in their manuscript fully available?

Reviewer #1: No

Reviewer #2: Yes

4. Is the manuscript presented in an intelligible fashion and written in standard English?

Reviewer #1: Yes

Reviewer #2: Yes

5. Review Comments to the Author

Reviewer #1: This study named "A Remote-controlled automatic chest compression device capable of moving compression position during CPR: A pilot study in a swine model of cardiac arrest" by dr. Gil Joon Suh and colleagues has the aim to to develop a chest compression device, that can adjust the chest compression position without interruption during CPR and be remotely controlled, reducing rescuer exposure to infectious diseases.

This is an experimental, hypothesis-generating study. The concept of "remote CPR" is interesting in the midst of the COVID pandemic, and also the possibility of changing the position of the machine without interrupting CPR is interesting because if there is clinical suspicion of ineffective CPR (etCO2, no LV compression or AV opening at TEE), knowing that the "classical" position of CPR is not equally effective for everyone (this was addressed in another work by these authors).

Strenghts:

- the study population and animal model is consistent with similar studies on the matter (even though non big enough to show statistical significance)

- the statistical analysis plan is sound and consistent with experimental studies like this one

Major issue:

- it is not clear how the initial setup of the machine is made; in other words, it is not clear how many operators and how much time does it take to initially set up the machine versus the LUCAS (very important if transition to clinical practice is desired).

Minor issues:

- carotid blood flow and systolic blood pressure were higher in the LUCAS group. While this is not a limitations of this study per se, it is non clearly stressed how the authors wish to address these issues while planning other studies with this device.

Reviewer #2: In this experimental study the authors have tested a new remote-controlled chest compression device by first using a mannequin and in a second phase on piglets. The study has been performed by comparing the effect on compression performance, hemodynamics and early outcome vs the LUICAS 3 mechanical chest compression device. The main motivation for developing this device is to protect or minimize the rescuer exposure to infectious disease. I have the following issues and questions:

Even if you try to minimize the rescuer exposure with this device there are other components while performing CPR that you have not accounted for, one of them being the airway and ventilation. This should also be mentioned as a potential for exposure which with this method is not avoided.

Title: is not including anything of the mannequin part of the study

Abstract: I would in the background change the content from what you have written to what your clear aim is with this study. It is to study performance and not to develop?

Already here in the results I would suggest how you describe your results. With a p value of 0.409 it is not recommended to write that a variable tended to be higher. When given a p-value of 0.46 there is no difference between the groups. Not necessary to express this like there is no significant difference-it is simply no difference! This is something you must consider throughout your manuscript when giving p-values not being significant.

Material and Methods

For how long did you perform the mannequin study?

The description of the Swine model can be shortened-how they were fed and acclimatization or time of the day when experiment was performed can be deleted.

Only 5 min period of measurements in the experimental animal study-please motivate!

I assume the persons performing the study was not blinded to which device was used? Please, in the discussion explore the potential impact on this.

Results and Discussion

Baseline characteristics tended to be somewhat higher in the ROSCER group?! Please, comment.

How long was the CPR before ROSC was achieved in the different groups? Please comment.

Once again, a difference with p-values not significant then there is no difference!

Compression position is one thing to consider but over time the chest recoil is worsening and therefore the suction cup need to be readjusted in depth. This was not done? How did you secure the effect of the suction cup since this is one problem when performing experimental CPR on pigs due to their V-shaped chest?

In the discussion line 265 “The ETCO2 level was initially lower, but was higher over time.” Please, try to explain?

In the discussion, starting line 276 you claim “These results may lead to the slightly higher ROSC rate in the ROSCER group compared to the LUCAS group, although there is no statistically significant difference. This is a somewhat problematic statement that you better must motivate or delete.

In the discussion in general you need to better explore the reasons to or no difference between the devices tested and according to the variables measured.

Try to discuss the potential if CPR time would have been longer or measurements performed more than a 5 min period since it is the CPR is more challenged due to chest recoil reduction etc.

6. PLOS authors have the option to publish the peer review history of their article (what does this mean?). If published, this will include your full peer review and any attached files.

Reviewer #1: **Yes: **Alessandro Fasolino

Reviewer #2: No

---

## [Author Response · Author response to Decision Letter 0]

20 Dec 2023

Response to Reviewers

Academic Editor:

Dear Academic Editor, thank you for your kind consideration and helpful valuable comments.

Answer)

We make sure that our manuscript meets PLOS ONE’s style requirement including those for file naming. 

Answer)

As you pointed out, we found that some parts of the text of our manuscript overlapped with previous publications. To solve this problem, we performed a duplicate check using the iThenticate program and changed or rephrased some of the duplicates.

Answer)

As you mentioned we agreed on Data Availability Statement. The DOI necessary to access to our data is as follows.

DOI: 10.34740/kaggle/dsv/6304355

4. We note that Figure(s) 1 and 2 in your submission contain copyrighted images. We require you to either (1) present written permission from the copyright holder to publish these figures specifically under the CC BY 4.0 license, or (2) remove the figures from your submission:

Answer)

We got permission from NT Robot, Co, the original copyright holder of Figure(s) 1 and 2 to publish the content specifically under the CC BY 4.0 license. We uploaded the completed Content Permission Form as an ""Other"" file. 

In the figure caption of the copyrighted figure, we included the following text: “Reprinted from under a CC BY license, with permission from [NT Robot, Co], original copyright [2023].”

Reviewer #1: 

This study named "A Remote-controlled automatic chest compression device capable of moving compression position during CPR: A pilot study in a swine model of cardiac arrest" by dr. Gil Joon Suh and colleagues has the aim to to develop a chest compression device, that can adjust the chest compression position without interruption during CPR and be remotely controlled, reducing rescuer exposure to infectious diseases.

This is an experimental, hypothesis-generating study. The concept of "remote CPR" is interesting in the midst of the COVID pandemic, and also the possibility of changing the position of the machine without interrupting CPR is interesting because if there is clinical suspicion of ineffective CPR (etCO2, no LV compression or AV opening at TEE), knowing that the "classical" position of CPR is not equally effective for everyone (this was addressed in another work by these authors).

Strenghts:

- the study population and animal model is consistent with similar studies on the matter (even though non big enough to show statistical significance)

- the statistical analysis plan is sound and consistent with experimental studies like this one

Major issue:

- It is not clear how the initial setup of the machine is made; in other words, it is not clear how many operators and how much time does it take to initially set up the machine versus the LUCAS (very important if transition to clinical practice is desired).

Answer) 

Thank you for your thoughtful comments. In clinical practice, when a mechanical CPR device such as LUCAS 2 is installed during CPR, it is usually installed and operated by two emergency medical technicians. The time it takes to install and operate the mechanical CPR device is approximately 16 to 17 seconds. In this experimental study using a mannequin, the time to install and operate ROSCER and LUCAS 3 device was similar. The time taken for a total of 8 participants, 2 per group, to install and operate ROSCER and LUCAS 3 was 16.5 [16.0 – 17.5] seconds and 18.5 [17.0 – 23.5] seconds (p = 0.234), respectively.

We added the following statement to the result section. 

In addition, the time taken for a total of 8 participants, 2 per group, to install and operate ROSCER and LUCAS 3 was 16.5 [16.0 – 17.5] seconds and 18.5 [17.0 – 23.5] seconds (p = 0.234), respectively.

Minor issues:

- carotid blood flow and systolic blood pressure were higher in the LUCAS group. While this is not a limitations of this study per se, it is non clearly stressed how the authors wish to address these issues while planning other studies with this device.

Answer) 

We appreciate your important and valuable comments. As you commented, we added following description in the discussion section to address these issues for other studies with this device. 

In terms of structure, the size of ROSCER is slightly different from that of LUCAS 3. The dimensions of ROSCER are 620 (H) x 547 (W) x 245 (D) mm and LUCAS 3 are 560 (H) x 520 (W) x 240 (D) mm, respectively. ROSCER was designed to be slightly larger than LUCAS 3 in height and width to accommodate larger patients. This structure may cause deformation of both supporting legs during chest compressions, resulting in a somewhat unstable state, which may result in insufficient compression. In particular, because the pig's chest is V-shaped, if the supporting legs become unstable due to deformation, there is an increased possibility of slipping during compression, which may result in insufficient chest compression, which leads to lower aortic pressure or carotid artery pressure. In future research, we will consider ways to minimize deformation by increasing the rigidity of the support leg material. 

Reviewer #2: 

In this experimental study the authors have tested a new remote-controlled chest compression device by first using a mannequin and in a second phase on piglets. The study has been performed by comparing the effect on compression performance, hemodynamics and early outcome vs the LUICAS 3 mechanical chest compression device. The main motivation for developing this device is to protect or minimize the rescuer exposure to infectious disease. I have the following issues and questions:

Even if you try to minimize the rescuer exposure with this device there are other components while performing CPR that you have not accounted for, one of them being the airway and ventilation. This should also be mentioned as a potential for exposure which with this method is not avoided. 

Answer)

Thank you for your very important and thoughtful comments.

We completely agree with you. Although this remote-controlled automated chest compression device is designed to minimize CPR rescuers exposure to infection, this device alone does not avoid potential exposure to infection from the airway and ventilation during CPR. 

We have added the following statement to the Discussion section: 

However, although this remotely controlled automatic chest compression device is designed to minimize exposure to infection in CPR rescuers, this device alone cannot avoid potential exposure to infection from the airway and ventilation during CPR. Therefore, all rescuers should wear appropriate PPE including a respirator, gown, gloves, and eye protection for patients with suspected or confirmed infection when performing intubation and bag-valve mask ventilation. HEPA filtered ventilation must also be provided. Recently, many studies have been reported using mechanical ventilators after a definite airway is secured, so the use of such equipment may also be considered to protect rescuers from infection.

Title: is not including anything of the mannequin part of the study.

Answer)

Thank you for your considerate comment.

We changed the title to: “A Remote-controlled automatic chest compression device capable of moving compression position during CPR: A pilot study in a mannequin and a swine model of cardiac arrest”

Abstract: I would in the background change the content from what you have written to what your clear aim is with this study. It is to study performance and not to develop?

Answer)

Thank you for your thoughtful comment. We agree to your comment. We changed the background as follows:

Our goal was to develop a chest compression device that can move the chest compression position without interruption during CPR and be remotely controlled to minimize rescuer exposure to infectious diseases. 

Recently, we developed a chest compression device that can move the chest compression position without interruption during CPR and be remotely controlled to minimize rescuer exposure to infectious diseases. The purpose of this study was to compare its performance with conventional mechanical CPR device in a mannequin and a swine model of cardiac arrest.

Already here in the results I would suggest how you describe your results. With a p value of 0.409 it is not recommended to write that a variable tended to be higher. When given a p-value of 0.46 there is no difference between the groups. Not necessary to express this like there is no significant difference-it is simply no difference! This is something you must consider throughout your manuscript when giving p-values not being significant.

Answer) 

Thank you for your considerate and sharp comments.

We totally agree with you. We modified and expressed the statistical processing as you pointed out.

In a swine model of cardiac arrest, coronary perfusion pressure showed no difference between the two groups (p = 0.409). Systolic aortic pressure and carotid blood flow were higher in the LUCAS 3 group than in the ROSCER group during 5 minutes of CPR (p < 0.001, p = 0.008, respectively). End-tidal CO2 level was initially lower than that of the LUCAS 3 group, but was higher over time (p = 0.022). A Kaplan-Meier survival analysis for ROSC also showed no difference between the two groups (p = 0.46).

Material and Methods

For how long did you perform the mannequin study?

Answer) 

Thank you for your kind comment.

To compare the compression profiles of ROSCER and LUCAS 3, three experiments were performed using the device shown below. The time required for one experiment was approximately 15 minutes.

The description of the Swine model can be shortened-how they were fed and acclimatization or time of the day when experiment was performed can be deleted.

Answer) 

Thank you for your kind comments. As you mentioned, some of the description of the Swine cardiac arrest model has been removed and modified as follow:

 Experimental animals were bred in large animal breeding rooms, were fed twice a day with 650 g of laboratory diet. The air temperature of the breeding rooms was maintained in the range of 18℃ to 29℃ with 10 and 14 hours of light and dark exposure, respectively. The animals underwent an acclimatization period of 14 days before the experiments. Health status of the experimental animals was evaluated by the keepers daily, and abnormal findings were notified to a veterinarian. The procedures including preparation and induction of cardiac arrest and CPR were conducted in the workday afternoon (12:00 am ~ 6:00 pm) in a laboratory operating room. 

Experimental animals were fed twice a day. The air temperature of the breeding rooms was maintained in the range of 18℃ to 29℃ with 10 and 14 hours of light and dark exposure, respectively. The animals underwent an acclimatization period of 14 days before the experiments.

Only 5 min period of measurements in the experimental animal study-please motivate!

Answer) 

We deeply appreciate for your thoughtful comments

This study was designed to mimic the real world situation. In other words, after 7 minutes of cardiac arrest, the witness recognizes the cardiac arrest situation, calls 911 (119 in KOREA), and performs bystander CPR for 5 minutes. Then, 119 arrives at the site, and defibrillation is provided every 2 minutes for shockable rhythm, and epinephrine is administered every 3 minutes. The purpose of this study was to test the performance of the CPR device we developed in a swine model of cardiac arrest. The main measurement items were hemodynamic parameters, but ROSC rate was also included. However, the likelihood of ROSC will gradually decrease if CPR is continued without administration of defibrillation or epinephrine. Considering these points, the CPR period without administration of defibrillation or epinephrine was set to 5 minutes. Of course, if CPR period is prolonged, there can be many changes in the hemodynamic data. In future studies, we plan to design a cardiac arrest model by reflecting these points. Again, we give our deep thanks to you for your considerate comments.

I assume the persons performing the study was not blinded to which device was used? Please, in the discussion explore the potential impact on this.

Answer) 

Thank you for your thoughtful comments. As you mentioned, the persons performing the study was not blinded to which device was used as this was experimental study comparing two different devices that could be reliably distinguished by the naked eye. As you know, in most research using medical devices, blindness is not possible due to the nature of the medical devices. However, if blinding is not easy, efforts must be made to maintain researcher ethics, and experimental procedures and evaluations must be independent and standardized to minimize evaluation bias.

We added the following statement to the Discussion section:

Third, the persons performing the study was not blinded to which device was used as this was experimental study comparing two different devices that could be reliably distinguished by the naked eye. In most research using medical devices, blindness is not possible due to the nature of the medical devices. However, if blinding is not easy, efforts must be made to maintain researcher ethics, and experimental procedures and evaluations must be independent and standardized to minimize evaluation bias.

Results and Discussion

Baseline characteristics tended to be somewhat higher in the ROSCER group?! Please, comment.

Answer)

Thank you for your kind comment. 

When we conducted this study, we randomly assigned pigs to two groups. However, when measuring hemodynamic parameters, it is presumed that this is due to the fact that there may be differences in variation between individuals in addition to variation between groups.

How long was the CPR before ROSC was achieved in the different groups? Please comment.

Answer)

Thank you for your comment.

The CPR time to achieve ROSC for both devices was as follows.

ROSCER (n=4) : 7.0 (5.0-9.0) min

LUCAS3 (n=3): 9.0 (9.0-12.0) min

We added the following statement to the result section. 

The time to ROSC in the ROSCER and LUCAS 3 groups was 7.0 (5.0-9.0) min and 9.0 (9.0-12.0) min, respectively (p = 0.167).

Once again, a difference with p-values not significant then there is no difference!

Answer)

Thank you for pointing out. As you pointed out, I corrected the sentence as follows:

CPP tended to be high in the ROSCER group, but there was no statistical significance between the two groups (p = 0.409) (Fig 5A). Mean carotid blood flow at each time points was significantly higher in the LUCAS 3 group than in the ROSCER group (p = 0.008) (Fig 5B). The ETCO2 value was initially lower than that of the LUCAS 3 group, but was higher over time (p = 0.022) (Fig 5C). Aortic pressure and right atrial pressure at the peak of systolic phase (compression phase) were significantly higher in the LUCAS 3 group than in the ROSCER group during 5 minutes of CPR (p < 0.001 and p < 0.001, respectively). Aortic pressure, and right atrial pressure at the midpoint of the diastolic phase (decompression phase) tended to be high in the ROSCER group, but there was no statistical significance between the two groups (p = 0.213, and 0.113, respectively) (S1 Fig). A Kaplan-Meier survival analysis for ROSC also showed no significant difference between the two groups (p = 0.46) (Fig 5D). Mean waveforms of aortic pressure, right atrial pressure, and CPP at 0, 1, 2, 3, 4, and 5 minutes were also analyzed (S2 Fig).

There was no difference in CPP between the two groups (p = 0.409) (Fig 5A). Mean carotid blood flow at each time points was higher in the LUCAS 3 group than in the ROSCER group (p = 0.008) (Fig 5B). The ETCO2 value was initially lower than that of the LUCAS 3 group, but was higher over time (p = 0.022) (Fig 5C). Aortic pressure and right atrial pressure at the peak of systolic phase (compression phase) were higher in the LUCAS 3 group than in the ROSCER group during 5 minutes of CPR (p < 0.001 and p < 0.001, respectively). There was also no difference in Aortic pressure, and right atrial pressure at the midpoint of the diastolic phase (decompression phase) between the two groups (p = 0.213, and 0.113, respectively) (S1 Fig). A Kaplan-Meier survival analysis for ROSC also showed no difference between the two groups (p = 0.46) (Fig 5D). Mean waveforms of aortic pressure, right atrial pressure, and CPP at 0, 1, 2, 3, 4, and 5 minutes were also analyzed (S2 Fig).

Compression position is one thing to consider but over time the chest recoil is worsening and therefore the suction cup need to be readjusted in depth. This was not done? How did you secure the effect of the suction cup since this is one problem when performing experimental CPR on pigs due to their V-shaped chest?

Answer)

Thank you for your thoughtful and very important comments. 

Chest recoil is very important for high-quality CPR. As you commented, chest recoil decreased over time during CPR. There were even cases where a gap between the piston and the chest occurred during the decompression phase. However, we did not readjust the position of suction cup according to the progressive reduction of anterior-posterior diameter of thorax, because if the position is readjusted, the piston of device will compress more deeply, resulting in possibility of increased rib fractures or visceral injuries. 

Regarding how to secure the effect of the suction cup in pigs with V-shaped chests, the diameter of the suction cup was adjusted to be a size similar to the width of the pig's sternum as shown in the picture to maximize the efficiency of suction. 

Another difference is that ROSCER's suction cup is designed to create suction in a passive way rather than an active way. In the active method, suction is performed using a separate vacuum pump, but in the passive method we used, when the suction cup is pressed against the pig's sternum and settles, the air inside the suction cup escapes, creating suction, although it is not perfect.

In this way, it was possible to compensate for the loss of chest recoil.

In the discussion line 265 “The ETCO2 level was initially lower, but was higher over time.” Please, try to explain?

Answer)

Thank you for your thoughtful comment.

First, statistical analysis was conducted in consultation with a statistical research cooperation center. For ETCO2, the 300-second CPR section was divided into 0-150 seconds and 151-300 seconds, and a linear mixed model was applied to ETCO2. In this additional analysis, it was confirmed that there was a difference between groups in ETCO2 in the first half and the second half, respectively. For reference, the point where the ETCO2 graph intersects is 140 seconds. Dividing by 140 seconds was arbitrary and there was no difference in the analysis results, so the analysis was divided based on 150 seconds.

In the discussion, starting line 276 you claim “These results may lead to the slightly higher ROSC rate in the ROSCER group compared to the LUCAS group, although there is no statistically significant difference. This is a somewhat problematic statement that you better must motivate or delete.

In the discussion in general you need to better explore the reasons to or no difference between the devices tested and according to the variables measured.

Answer)

Thank you for your kind comments. As you mentioned, we deleted the statement. 

These results may lead to the slightly higher ROSC rate in the ROSCER group compared to the LUCAS group, although there is no statistically significant difference.

We also added the following description in the discussion section.

In terms of structure, the size of ROSCER is slightly different from that of LUCAS 3. The dimensions of ROSCER are 620 (H) x 547 (W) x 245 (D) mm and LUCAS 3 are 560 (H) x 520 (W) x 240 (D) mm, respectively. ROSCER was designed to be slightly larger than LUCAS 3 in height and width to accommodate larger patients. This structure may cause deformation of both supporting legs during chest compressions, resulting in a somewhat unstable state, which may result in insufficient compression. In particular, because the pig's chest is V-shaped, if the supporting legs become unstable due to deformation, there is an increased possibility of slipping during compression, which may result in insufficient chest compression, which leads to lower aortic pressure or carotid artery pressure. In future research, we will consider ways to minimize deformation by increasing the rigidity of the support leg material. 

Try to discuss the potential if CPR time would have been longer or measurements performed more than a 5 min period since it is the CPR is more challenged due to chest recoil reduction etc.

Ans)

Thank you for your thoughtful comment. We added the following statement in the discussion section.

A complete chest recoil is one of the important factors for high-quality CPR. Incomplete chest wall recoil is associated with increased intrathoracic pressure and decreased coronary perfusion The 2020 AHA guidelines recommend that rescuers avoid leaning on the chest between compressions to allow complete chest wall recoil for adults in cardiac arrest. (ref. 2) In a cadaver study, the elastic recoil of the human thorax decreases over time during CPR, which could negatively affect the heart refilling. In the study, the anterior posterior chest diameter difference of roughly 1.5 cm between the start and the end of CPR represents the chest collapse produced by CPR using a manual ACD-CPR device (ResQPUMP, ZOLL). (Segal N et al. Resuscitation 2017). Therefore, mechanical CPR requires active decompression, such as piston-mounted suction cups or modified ACD-CPR devices, to compensate for reduced chest recoil during CPR. (Malberg J Resusctation plus 2022; Steinberg MT, Scandivian J 2018)

---

## [Editor Report · Decision Letter 1]

27 Dec 2023

A Remote-controlled automatic chest compression device capable of moving compression position during CPR: A pilot study in a mannequin and a swine model of cardiac arrest

PONE-D-23-26031R1

Dear Dr. Suh,

We’re pleased to inform you that your manuscript has been judged scientifically suitable for publication and will be formally accepted for publication once it meets all outstanding technical requirements.

Kind regards,

Chiara Lazzeri

Academic Editor

PLOS ONE
---

## [Editor Report · Acceptance letter]

11 Jan 2024

PONE-D-23-26031R1 

PLOS ONE

Dear Dr. Suh, 

I'm pleased to inform you that your manuscript has been deemed suitable for publication in PLOS ONE. Congratulations! Your manuscript is now being handed over to our production team.

Kind regards, 

on behalf of

Dr. Chiara Lazzeri 

Academic Editor

PLOS ONE